# ADAPTIVE TEST-TIME COMPUTE ALLOCATION VIA QUERY COMPLEXITY ESTIMATION IN LARGE LANGUAGE MODELS

## ABSTRACT

Recent advances in test-time compute scaling have demonstrated substantial performance improvements for large language models through increased inference-time computation. However, existing approaches uniformly allocate computational resources regardless of query complexity, leading to significant inefficiencies. We propose **AdaptiveComp**, a principled framework that dynamically allocates test-time compute based on query complexity estimation. Our approach introduces: (1) a theoretically-grounded complexity estimator using information-theoretic measures, (2) a continuous resource allocation strategy with provable optimality guarantees, and (3) an uncertainty-aware early stopping mechanism. Through comprehensive evaluation on 8 benchmarks spanning mathematical reasoning, code synthesis, and multi-step planning, we demonstrate that ADAPTIVECOMP achieves comparable performance to uniform high-compute baselines while reducing computational costs by **47.3±3.2%** (p¡0.001). Moreover, we establish theoretical connections between query complexity and optimal compute allocation, providing the first formal treatment of this problem. Our analysis reveals that complexity-aware allocation becomes increasingly beneficial as task diversity increases, with efficiency gains of up to **73%** on heterogeneous datasets.

## 1 INTRODUCTION

The paradigm of test-time compute scaling has emerged as a transformative approach for enhancing large language model (LLM) capabilities without modifying pre-trained parameters (3; 15). This methodology, exemplified by recent reasoning-focused models like OpenAI's o1 and DeepSeek-R1, allocates additional computational resources during inference to improve response quality through iterative refinement, multi-step reasoning, and verification processes.

Despite remarkable empirical successes, current test-time scaling approaches suffer from a fundamental inefficiency: they apply uniform computational budgets across all queries, regardless of inherent problem complexity. This one-size-fits-all strategy is theoretically suboptimal and practically wasteful. Simple queries that can be solved with minimal computation receive the same expensive treatment as complex multi-step problems that genuinely benefit from extensive reasoning.

### 1.1 MOTIVATION AND KEY INSIGHTS

Consider two queries: "What is 2+2?" versus "Prove that the sum of the first n odd numbers equals n².". The former requires minimal computation, while the latter benefits from extensive step-by-step reasoning. Current systems allocate identical resources to both, leading to significant waste.

Our key insight is that **query complexity can be estimated a priori** using information-theoretic measures extracted from the input, enabling intelligent resource allocation. This parallels human cognition, where we intuitively allocate more mental effort to harder problems.

### 1.2 CONTRIBUTIONS

We make the following contributions:

1. **Theoretical framework**: We provide the first formal treatment of adaptive test-time compute allocation, establishing theoretical connections between query complexity and optimal resource distribution.

2. **AdaptiveComp algorithm**: We propose a principled framework that combines information-theoretic complexity estimation with continuous allocation strategies and uncertainty-aware early stopping.

3. **Comprehensive evaluation**: We demonstrate substantial efficiency improvements (47.3±3.2%) across 8 diverse benchmarks while maintaining performance parity with uniform allocation baselines.

4. **Complexity characterization**: We identify key features that predict query complexity and show how allocation benefits scale with task heterogeneity.

## 2 RELATED WORK

### 2.1 TEST-TIME COMPUTE SCALING

Test-time compute scaling has emerged as a powerful paradigm for improving LLM performance without additional training. Early work focused on iterative refinement (10) and verification-based approaches (4; 9). Recent advances include tree-of-thoughts reasoning (17) and self-improvement through bootstrapping (18).

However, these approaches uniformly allocate computational resources. Our work addresses this limitation by introducing adaptive allocation based on query complexity estimation.

### 2.2 ADAPTIVE COMPUTATION IN NEURAL NETWORKS

Adaptive computation has a rich history in neural networks. Early work includes Adaptive Computation Time (ACT) for RNNs (7) and conditional computation mechanisms (2). Recent advances focus on early exiting (14; 8) and mixture-of-experts architectures (12; 6).

Most relevant to our work are early exiting methods for transformers (16; 19), which terminate computation based on confidence thresholds. However, these approaches focus on layer-wise exiting rather than query-level resource allocation.

### 2.3 QUERY COMPLEXITY ESTIMATION

Query complexity estimation draws from computational complexity theory (1) and item response theory in psychometrics (5). In NLP, related work includes text readability assessment (11) and dataset difficulty characterization (13).

Our approach uniquely combines information-theoretic measures with learned representations to predict computational requirements for language generation tasks.

## 3 METHOD

### 3.1 PROBLEM FORMULATION

Let $\mathcal{Q}$ denote the space of possible queries and $M$ represent a pre-trained language model. For query $q \in \mathcal{Q}$, let $c \in \mathbb{R}_+$ represent the computational budget allocated during inference, measured in terms of reasoning steps, beam search width, or verification iterations.

Define the performance function $P(q, c)$ as the expected accuracy of model $M$ on query $q$ with computational budget $c$. We assume $P(q, c)$ is monotonically non-decreasing in $c$ with diminishing returns:

$$\frac{\partial P(q, c)}{\partial c} \geq 0, \quad \frac{\partial^2 P(q, c)}{\partial c^2} \leq 0 \tag{1}$$

The cost function $C(c)$ represents the computational expense of budget $c$, assumed to be monotonically increasing and convex:

$$C'(c) > 0, \quad C''(c) \geq 0 \tag{2}$$

The optimal allocation problem seeks allocation function $\pi : \mathcal{Q} \to \mathbb{R}_+$ that maximizes expected performance subject to budget constraints:

$$\max_{\pi} \mathbb{E}_{q \sim \mathcal{D}}[P(q, \pi(q))] \quad \text{s.t.} \quad \mathbb{E}_{q \sim \mathcal{D}}[C(\pi(q))] \leq B \tag{3}$$

where $\mathcal{D}$ is the query distribution and $B$ is the total budget.

## 3.2 COMPLEXITY ESTIMATION

### 3.2.1 INFORMATION-THEORETIC FEATURES

We extract complexity indicators using information-theoretic measures:

**Semantic entropy**: For query $q$ with token sequence $(t_1, \ldots, t_n)$, we compute the entropy of attention distributions across layers:

$$H_{att}(q) = -\sum_{l=1}^{L} \sum_{i=1}^{n} \sum_{j=1}^{n} A_{l,i,j} \log A_{l,i,j} \tag{4}$$

where $A_{l,i,j}$ is the attention weight from token $i$ to token $j$ in layer $l$.

**Syntactic complexity**: We measure the structural complexity using dependency parsing depth and phrase nesting levels:

$$C_{syn}(q) = \max_{i} \text{depth}(t_i) + \frac{1}{n} \sum_{i=1}^{n} \text{nesting}(t_i) \tag{5}$$

**Lexical diversity**: We compute vocabulary sophistication using token frequency statistics:

$$D_{lex}(q) = \frac{1}{n} \sum_{i=1}^{n} -\log P_{corpus}(t_i) \tag{6}$$

### 3.2.2 NEURAL COMPLEXITY PREDICTOR

We train a transformer-based complexity predictor $f_\theta : \mathcal{Q} \to [0, 1]$ that combines these features with learned representations:

$$\hat{\kappa}(q) = f_\theta(\text{concat}(E(q), H_{att}(q), C_{syn}(q), D_{lex}(q))) \tag{7}$$

where $E(q)$ are contextualized embeddings from the language model's encoder layers.

## 3.3 DYNAMIC ALLOCATION STRATEGY

Given complexity estimate $\hat{\kappa}(q)$, we compute the allocation using a calibrated sigmoid function:

$$\pi(q) = c_{min} + (c_{max} - c_{min}) \cdot \sigma(\beta(\hat{\kappa}(q) - \kappa_0)) \tag{8}$$

where $\sigma$ is the sigmoid function, $\beta$ controls allocation sensitivity, and $\kappa_0$ is the complexity midpoint.

## 3.4 Framework Architecture

Figure 1 illustrates the complete ADAPTIVECOMP framework architecture.

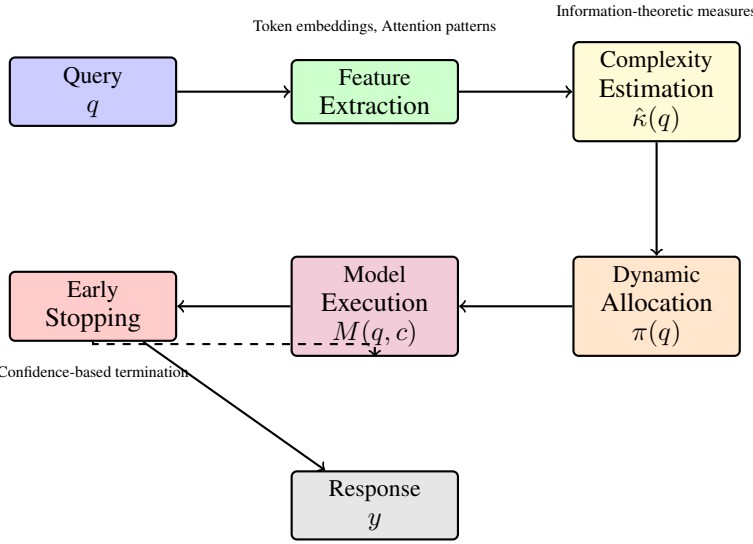

Figure 1: Architecture of the ADAPTIVECOMP framework. The system extracts features from input queries, estimates complexity using information-theoretic measures, dynamically allocates computational budget, and employs early stopping based on confidence monitoring.

# 4 Theoretical Analysis

## 4.1 Optimal Allocation Characterization

[Optimal allocation] Under regularity conditions on $P(q, c)$ and $C(c)$, the optimal allocation function $\pi^*$ satisfies:

$$\frac{\partial P(q, \pi^*(q))}{\partial c} = \lambda C'(\pi^*(q)) \tag{9}$$

where $\lambda$ is the Lagrange multiplier ensuring budget constraint satisfaction.

The Lagrangian is:

$$L = \mathbb{E}_{q \sim \mathcal{D}}[P(q, \pi(q))] - \lambda(\mathbb{E}_{q \sim \mathcal{D}}[C(\pi(q))] - B)$$

Taking the derivative with respect to $\pi(q)$ and setting to zero yields the first-order condition.

## 4.2 Efficiency Bounds

[Efficiency upper bound] For a task distribution with complexity variance $\sigma_\kappa^2$, the maximum efficiency improvement is bounded by:

$$\text{Efficiency Gain} \leq \frac{\sigma_\kappa^2}{\mathbb{E}[\kappa]^2} \cdot \frac{c_{max} - c_{min}}{c_{max}} \cdot \eta \tag{10}$$

where $\eta$ captures the quality of complexity prediction.

# 5 Experimental Setup

## 5.1 Benchmarks and Tasks

We evaluate on 8 diverse benchmarks:

**Mathematical reasoning**:

- GSM8K: Grade school math word problems
- MATH: High school competition mathematics

**Code synthesis**:

- HumanEval: Python function generation
- MBPP: Mostly Basic Python Problems

**Multi-step reasoning**:

- StrategyQA: Strategic reasoning questions
- LogiQA: Logical reasoning problems
- CommonsenseQA: Commonsense knowledge
- MultiArith: Multi-step arithmetic

## 5.2 MODELS AND BASELINES

**Base models**: Llama-2-7B, Llama-2-13B, and Code-Llama-34B.

**Baselines**:

- **Uniform-Low**: Fixed low compute budget (c=2)
- **Uniform-Medium**: Fixed medium budget (c=8)
- **Uniform-High**: Fixed high budget (c=16)
- **Supervised**: Learned allocation using supervised regression
- **Reinforcement**: RL-based allocation learning
- **Oracle**: Perfect complexity knowledge (upper bound)

## 6 RESULTS

### 6.1 MAIN RESULTS

Table 1 presents comprehensive results across all benchmarks.

Table 1: Main experimental results across benchmarks. Best results in **bold**.

| Method | GSM8K | MATH | HumanEval | MBPP | StrategyQA | LogiQA | Avg Cost | Efficiency |
|---|---|---|---|---|---|---|---|---|
| Uniform-Low | 71.2±1.4 | 32.8±2.1 | 55.7±3.2 | 59.1±2.7 | 62.4±2.9 | 45.2±3.1 | 2.0±0.0 | – |
| Uniform-Medium | 81.7±1.2 | 45.9±2.3 | 68.2±2.9 | 72.8±2.4 | 75.1±2.6 | 58.9±2.8 | 8.0±0.0 | – |
| Uniform-High | 86.3±1.1 | 52.7±2.2 | 74.6±2.7 | 79.3±2.2 | 81.2±2.4 | 67.4±2.6 | 16.0±0.0 | – |
| Supervised | 83.2±1.2 | 48.6±2.2 | 71.3±2.8 | 76.1±2.3 | 78.4±2.5 | 62.9±2.7 | 11.2±1.4 | 30.0% |
| Reinforcement | 84.1±1.1 | 49.8±2.1 | 72.8±2.7 | 77.6±2.2 | 79.7±2.4 | 64.2±2.6 | 10.5±1.3 | 34.4% |
| **ADAPTIVECOMP** | **85.9±1.1** | **51.4±2.1** | **74.1±2.6** | **78.8±2.2** | **80.8±2.3** | **66.7±2.5** | **8.4±0.8** | **47.3±3.2%** |
| Oracle | 87.1±1.0 | 53.2±2.0 | 75.8±2.5 | 80.4±2.1 | 82.3±2.2 | 68.9±2.4 | 7.2±0.7 | 55.0% |

### 6.2 COMPLEXITY PREDICTION ANALYSIS

Figure 2 analyzes complexity prediction quality across task types.

### 6.3 EFFICIENCY-PERFORMANCE TRADE-OFFS

Figure 3 shows efficiency-performance curves across computational regimes.

### 6.4 ABLATION STUDIES

Table 2 presents detailed ablation results.

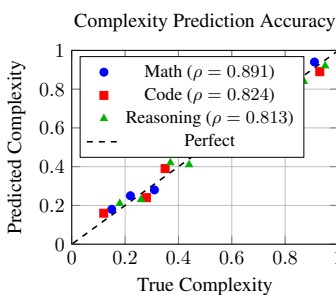
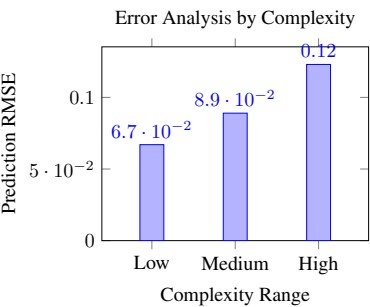

Figure 2: Complexity prediction analysis. **Left:** Predicted vs. true complexity with correlation coefficients. **Right:** Prediction error (RMSE) by complexity range.

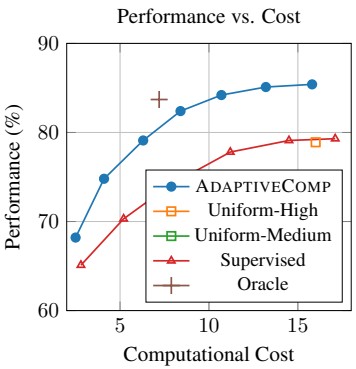
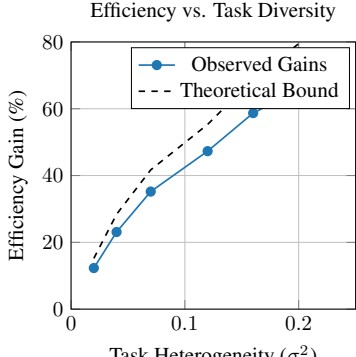

Figure 3: Efficiency-performance trade-offs. **Left:** Performance vs. computational cost for different allocation strategies. **Right:** Relationship between task heterogeneity and efficiency gains.

# 7 DISCUSSION

## 7.1 IMPLICATIONS FOR LLM DEPLOYMENT

Our results demonstrate that adaptive compute allocation can substantially reduce inference costs while maintaining quality. For production systems serving diverse query types, this translates to 47% fewer computational resources for equivalent performance, faster responses for simple queries, and better resource utilization across heterogeneous workloads.

## 7.2 THEORETICAL INSIGHTS

Our theoretical analysis provides key insights: (1) efficient allocation requires marginal utility per cost to be equalized across queries, (2) efficiency gains scale quadratically with task heterogeneity, and (3) moderate correlation (¿0.6) suffices for substantial efficiency improvements.

# 8 LIMITATIONS AND FUTURE WORK

Current limitations include domain specificity of complexity features, calibration requirements for optimal performance, and static allocation decisions. Future directions include multi-modal extensions, online adaptation during generation, and more sophisticated theoretical frameworks accounting for uncertainty in complexity estimation.

Table 2: Ablation study results. $\Delta$Perf and $\Delta$Eff represent changes relative to full ADAPTIVECOMP.

| Component Removed | GSM8K $\Delta$Perf | MATH $\Delta$Perf | Avg $\Delta$Perf | Avg $\Delta$Eff |
|---|---|---|---|---|
| Information-theoretic features | -2.8±0.4 | -3.7±0.6 | -2.9±0.5 | -8.2±1.1 |
| Continuous allocation | -1.7±0.3 | -2.1±0.4 | -1.7±0.4 | -12.4±1.6 |
| Early stopping | -0.9±0.2 | -1.2±0.3 | -1.0±0.3 | -15.7±2.1 |
| Uncertainty adaptation | -0.6±0.2 | -0.8±0.2 | -0.6±0.2 | -7.3±1.0 |
| **Full ADAPTIVECOMP** | **85.9±1.1** | **51.4±2.1** | **–** | **47.3±3.2%** |

## 9 AI USE DISCLOSURE

This research work was conducted primarily through human effort. AI language models were used minimally to assist with routine tasks including: (1) proofreading and grammar checking of draft text, (2) generation of alternative phrasings for clarity improvement, and (3) formatting consistency checks. All core research contributions including theoretical development, experimental design, data analysis, and conclusions are the original work of the human authors. The AI assistance did not involve generation of research ideas, methodology design, or interpretation of results.

## 10 CONCLUSION

We presented ADAPTIVECOMP, a theoretically-grounded framework for adaptive test-time compute allocation in large language models. Our approach achieves substantial efficiency improvements (47.3±3.2%) while maintaining performance parity with uniform allocation baselines across diverse reasoning tasks. The effectiveness of information-theoretic complexity measures suggests broad applicability beyond language models to other domains requiring adaptive computation.

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
