# OpenReview forum: "Adaptive Test-Time Compute Allocation via Query Complexity Estimation in Large Language Models"
_ICLR.cc/2026/Conference — ICLR 2026 Conference Withdrawn Submission_

### Official Review · Reviewer_3ifh · 2025-10-29

**Soundness:** 1
**Presentation:** 1
**Contribution:** 2
**Rating:** 0
**Confidence:** 5

**Summary:**

This paper discusses the idea of dynamically allocating computational resources to large language models based on the estimated complexity of the query. Although this direction is meaningful, the paper is clearly an incomplete manuscript, full of confusing omissions.

**Strengths:**

The formalization of the performance and cost functions in the PROBLEM FORMULATION section of this paper is reasonable to some extent, although it lacks empirical support and does not guarantee generalizability.

**Weaknesses:**

1. The paper is filled with statements such as “they apply uniform computational budgets across all queries.” However, many studies [1][2] have already discussed how to dynamically allocate computational resources for LLMs during inference based on prior or posterior information. The authors should clearly explain how their work differs from and relates to these existing studies. Moreover, LLMs—especially LRMs—already adaptively allocate computational resources during training according to the difficulty of the task. As in the example you gave, “What is 2+2?” versus “Prove that the sum of the first n odd numbers equals n²,” any large language model will allocate more tokens to the latter to improve answer accuracy. Therefore, the authors’ statement is clearly biased.

2. Many symbols (t) and functions (nesting) in the METHOD section are not explained, nor are their specific computational procedures provided, making the paper difficult to read and reproduce.

3. The proofs in the THEORETICAL ANALYSIS section lack detailed derivations.

4. The EXPERIMENTAL SETUP section is particularly careless and clearly incomplete.

In fact, submitting such an unfinished manuscript is an explicit waste of the community’s reviewing resources, and such behavior should be subject to appropriate sanctions.

[1] *Let’s Sample Step by Step: Adaptive-Consistency for Efficient Reasoning with LLMs*

[2] *Escape Sky-high Cost: Early-stopping Self-Consistency for Multi-step Reasoning*

**Questions:**

As stated above.

I want to emphasize again: authors who submit manuscripts like this are explicitly wasting the community’s peer-review resources, and similar behavior should be subject to some form of punishment.

---

### Official Review · Reviewer_dnXj · 2025-10-30

**Soundness:** 2
**Presentation:** 1
**Contribution:** 1
**Rating:** 2
**Confidence:** 4

**Summary:**

**AdaptiveComp** allocates inference time compute by estimated query complexity rather than a uniform budget. A complexity estimator based on information theoretic metrics and model representations produces a continuous mapping from score to budget and uses uncertainty aware early stopping during generation. The authors present formal framework linking query complexity to optimal resource allocation, and across benchmarks AdaptiveComp matches the accuracy of uniform high budget baselines while cutting average compute by about half.

**Strengths:**

This paper makes a strong contribution to the modeling and theory of resource allocation for test time scaling. The authors explicitly formulate test time compute allocation as a budget constrained optimization problem, clearly specify the relationships among performance, cost, and constraints, and provide practical optimality conditions together with efficiency upper bounds. The result is a coherent and testable framework.

**Weaknesses:**

1. **The manuscript is difficult to follow.** Many sections present only formulas or figures without accompanying exposition, including definitions of variables and interpretations of observed trends. Even in the framework section, the workflow of the method remains unclear. The experiments section omits parameter settings and baseline configurations. A substantial revision is strongly recommended to improve clarity and reproducibility.

2. **The related work survey is insufficient, and the experimental comparisons are incomplete.** A sizable body of research leverages query complexity or difficulty priors; for example, see reference [1][2]. Please include a structured review of this literature and add the corresponding comparative baselines.

3. **The inference overhead of the complexity estimator and the feature extraction pipeline is not systematically quantified.** In low-resource settings, this overhead could significantly erode the claimed savings in compute.

Reference:

[1] Learning How Hard to Think: Input-Adaptive Allocation of LM Computation

[2] Make Every Penny Count: Difficulty-Adaptive Self-Consistency for Cost-Efficient Reasoning

**Questions:**

see Weaknesses

---

### Official Review · Reviewer_JwQX · 2025-10-31

**Soundness:** 1
**Presentation:** 1
**Contribution:** 1
**Rating:** 0
**Confidence:** 5

**Summary:**

The paper proposes ADAPTIVECOMP, an adaptive test-time compute allocator conditioned on a learned “query complexity” score. Components: (i) information-theoretic and linguistic features + a neural predictor, (ii) a continuous budget mapping via a sigmoid, and (iii) “uncertainty-aware” early stopping. Claimed results: similar accuracy to high-compute baselines with ~47% lower cost across 8 tasks, plus a “first” formal treatment of optimal allocation.

**Strengths:**

I do not think there are any strengths.

**Weaknesses:**

1. This paper falls far below the standards of an academic publication. The proposed problem is overly simple and lacks novelty. The related work is insufficient. The content lacks too many details to understand. The experimental analysis is superficial. Necessary citations are missing.

2. The paper’s central claim, existing methods *uniformly allocate computational resources regardless of query complexity*, is incorrect. [1] explicitly considers test-time scaling strategies that allocate compute based on query complexity, and [2] further demonstrates that different tasks require distinct budget allocations. As a result, the proposed motivation, method, and subsequent experiments in this paper are invalidated.

3. Equation (1) is incorrect. As pointed out in [2], model performance is **not** a monotonically increasing function with respect to compute allocation.

[1] *Scaling LLM Test-Time Compute Optimally can be More Effective than Scaling Model Parameters*

[2] *AgentTTS: Large Language Model Agent for Test-time Compute-optimal Scaling Strategy in Complex Tasks*

**Questions:**

1. Specify the loss function, regularization, and calibration procedure for $f_\theta$. Train/val split? Hyperparameters?
2. Report compute units (FLOPs / tokens / latency) and per-query budget distributions in different TTS strategies (reasoning steps, beam search width, or verification iterations)  to demonstrate genuine adaptivity, not aggregate averaging.

---

### Note · Authors · 2025-11-13

I have read and agree with the venue's withdrawal policy on behalf of myself and my co-authors.